# Advanced glycation end-product crosslinking activates a type VI secretion system phospholipase effector protein

Steven J. Jensen[1,5], Bonnie J. Cuthbert[2,5], Fernando Garza-Sánchez[1], Colette C. Helou [2], Rodger de Miranda[2], Celia W. Goulding[2,3] & Christopher S. Hayes [1,4] ✉

Advanced glycation end-products (AGE) are a pervasive form of protein damage implicated in the pathogenesis of neurodegenerative disease, atherosclerosis and diabetes mellitus. Glycation is typically mediated by reactive dicarbonyl compounds that accumulate in all cells as toxic byproducts of glucose metabolism. Here, we show that AGE crosslinking is harnessed to activate an antibacterial phospholipase effector protein deployed by the type VI secretion system of *Enterobacter cloacae*. Endogenous methylglyoxal reacts with a specific arginine-lysine pair to tether the N- and C-terminal α-helices of the phospholipase domain. Substitutions at these positions abrogate both crosslinking and toxic phospholipase activity, but in vitro enzyme function can be restored with an engineered disulfide that covalently links the N- and C-termini. Thus, AGE crosslinking serves as a bona fide post-translation modification to stabilize phospholipase structure. Given the ubiquity of methylglyoxal in prokaryotic and eukaryotic cells, these findings suggest that glycation may be exploited more generally to stabilize other proteins. This alternative strategy to fortify tertiary structure could be particularly advantageous in the cytoplasm, where redox potentials preclude disulfide bond formation.

All bacteria compete for limited resources in the environment, with many species deploying specialized secretion systems to deliver toxic effector proteins directly into neighboring cells[1]. The type VI secretion system (T6SS) is a common mechanism of effector delivery utilized by many Gram-negative bacteria. T6SSs are dynamic multi-protein machines that function like contractile bacteriophages[2–4]. The T6SS duty cycle begins with the docking of a phage-like baseplate subassembly onto the membrane-embedded trans-envelope complex, which forms the export conduit across the inner and outer membranes of the bacterium[5,6]. The baseplate then serves as the origin for polymerization of an elongated contractile sheath that surrounds a central tube of Hcp hexamers[7]. The Hcp tube is capped with trimeric VgrG, which is structurally similar to phage tail-spike proteins[8]. Toxic effector proteins are packaged into the lumen of the Hcp tube and also carried by the VgrG spike[9–11]. Upon contraction of the sheath, the spike-tipped tube is ejected from the cell through the trans-envelope complex. Nearby bacteria are perforated by the projectile, allowing toxic effector proteins to be deposited directly into the target-cell periplasm.

Because antibacterial T6SS effectors are potentially auto-inhibitory, they are invariably encoded with cognate immunity proteins that

[1]Department of Molecular, Cellular and Developmental Biology, University of California, Santa Barbara, Santa Barbara 93106, USA. [2]Department of Molecular Biology & Biochemistry, University of California, Irvine, Irvine 92697, USA. [3]Department of Pharmaceutical Sciences, University of California, Irvine, Irvine 92697, USA. [4]Biomolecular Science and Engineering Program, University of California, Santa Barbara, Santa Barbara 93106, USA. [5]These authors contributed equally: Steven J. Jensen, Bonnie J. Cuthbert. ✉e-mail: hayescs@ucsb.edu

protect the cell from self-intoxication[1,2]. We recently reported that the T6SS lipase immunity (Tli) protein of *Enterobacter cloacae* is unusual in that it performs two distinct functions[12]. The mature Tli lipoprotein is secreted to the periplasmic space, where it acts as a canonical immunity factor to neutralize incoming T6SS lipase effector (Tle) proteins[12]. *E. cloacae* cells also synthesize cytoplasmic Tli from alternative translation initiation sites downstream of the secretion signal sequence. This cytosolic pool of immunity protein is required for the secretion of active Tle[12]. Given that immunity proteins bind directly to their cognate effectors[1], Tle activation presumably depends on transient interactions with cytoplasmic Tli before the lipase is packaged into the T6SS apparatus for export. Thus, Tli immunity protein has the paradoxical ability to activate its cognate effector protein through an unknown mechanism. Here, we show that Tle activation entails covalent modification with endogenous methylglyoxal to form an intramolecular crosslink that stabilizes the lipase domain.

## Results

### Modification of the Tle phospholipase domain

To explore the biochemical basis of lipase activation, we used immunoblotting to monitor Tle produced in the absence and presence of Tli. When placed under the control of a plasmid-borne arabinose-inducible promoter, the isolated *tle* gene produces a single Tle protein species (Fig. 1a, lane 3). By contrast, expression of the *tle-tli* gene pair yields two Tle species, including an apparently processed form that migrates more rapidly during gel electrophoresis (Fig. 1a, lane 4). These same two species of Tle accumulate when the *tle-tli* construct is induced in *E. coli* cells (Fig. 1a, lanes 7 & 8), indicating that processing does not require factors encoded by the T6SS locus of *E. cloacae*. Given that only about half of the Tle chains appear to be processed, we reasoned that the *tle-tli* construct may not produce enough cytoplasmic Tli to support complete effector conversion. Therefore, we deleted the secretion signal sequence from Tli (Δss-Tli) and placed the construct under the control of a strong isopropyl-β-D-thiogalactoside (IPTG) inducible

promoter. We also generated a new arabinose-inducible construct that produces Tle with an N-terminal His6 epitope to facilitate purification. Co-induction of these constructs produces fully converted His6-Tle (Supplementary Fig. 1a, lane 3). Mass spectrometry of the purified His6-Tle•Δss-Tli complex shows that the immunity protein has the expected mass (Supplementary Fig. 1b). However, His6-Tle exhibits a + 36 Da shift relative to the predicted mass (Supplementary Fig. 1b), indicating that the effector is not processed by proteases. We were unable to characterize unmodified His6-Tle, because this form is insoluble (Supplementary Fig. 1a, lane 5) and does not ionize well for mass spectrometry. We also examined the C-terminal lipase domain of Tle (corresponding to residues Thr172 - Ala472) and found that it is modified when produced with Δss-Tli (Fig. 1b, lane 1). The modified lipase domain can be isolated from the immunity protein by $Ni^{2+}$-affinity chromatography under denaturing conditions for comparison with the unmodified form (Fig. 1b, lanes 2 and 3). The modified domain exhibits the same +36 Da mass shift observed with full-length Tle (Fig. 1c), but we were unable to measure the mass of unmodified domain accurately. In vitro enzyme assays show that the modified lipase catalyzes phospholipase A1 and A2 reactions, which entail hydrolysis of phospholipid *sn*-1 and *sn*-2 acyl chains, respectively (Fig. 1d, e). Both activities are neutralized when the lipase domain is pre-incubated with purified Δss-Tli immunity protein (Fig. 1d, e). By contrast, unmodified lipase has no enzymatic activity in vitro (Fig. 1d, e). The modified form of full-length His6-Tle also exhibits phospholipase A1 and A2 activities, albeit at lower levels than the lipase domain construct (Supplementary Figs. 1c, d). To determine whether lipase activity is required for modification, we substituted the catalytic Ser341 residue[12] with Ala and produced the domain with Δss-Tli. The resulting Ser341Ala lipase is modified to the same extent as the wild-type domain (Fig. 1b, lanes 4 and 5), indicating that the catalytic center is not required for modification. As expected, the Ser341Ala lipase domain has no phospholipase activity regardless of modification status (Fig. 1d, e). Thus, the lipase domain of Tle is modified when

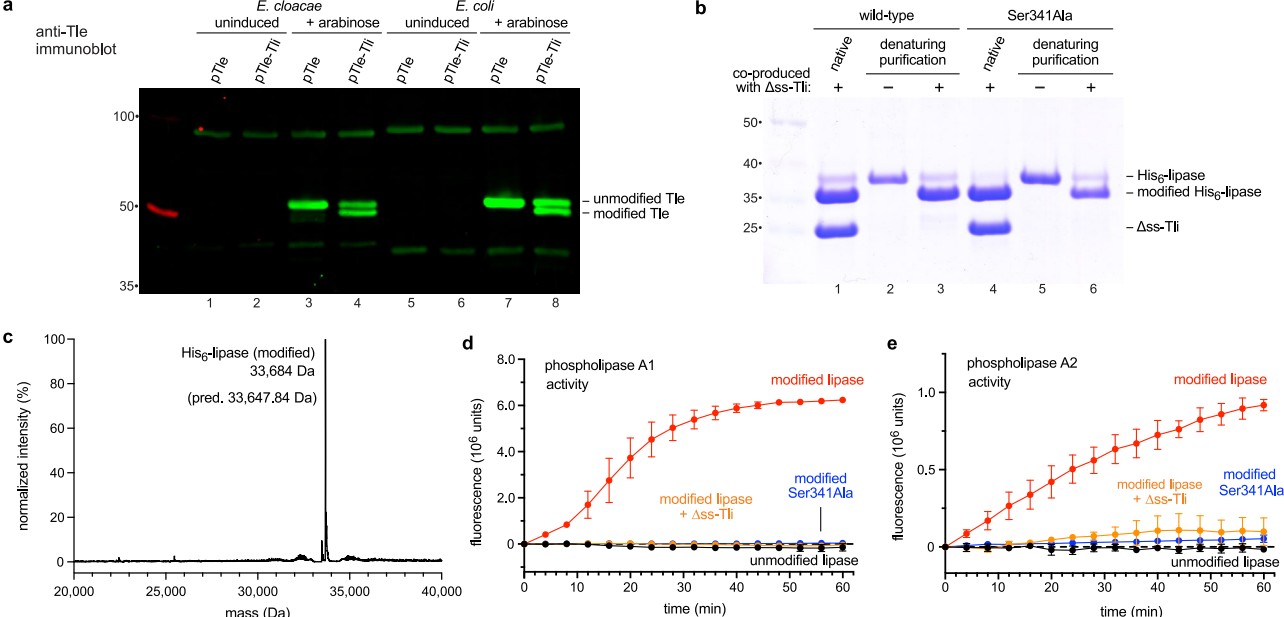

**Fig. 1 | Modification of the Tle phospholipase domain. a** Tle and Tle-Tli production was induced with arabinose in *E. cloacae* and *E. coli* cells. Urea-soluble total protein was extracted for immunoblot analysis using anti-Tle polyclonal antisera. Molecular mass markers (kDa) are on the left. This experiment was performed independently three times with similar results. **b** His6-lipase domains were produced with (or without) Δss-Tli in *E. coli* cells, then purified by $Ni^{2+}$-affinity chromatography under non-denaturing (native) or denaturing conditions for SDS-PAGE

analysis. Molecular mass markers (kDa) are on the left. **c** Deconvoluted mass spectrum of purified modified His6-lipase domain. **d** In vitro phospholipase A1 activities of purified lipase domains. **e** In vitro phospholipase A2 activities of purified lipase domains. Phospholipase activity data are presented as mean values ± standard deviation for three independent experiments. Source data are provided as a Source Data file.

**Table 1 | Data collection and refinement statistics for lipase•Δss-Tli crystal structures**

| | S-SAD | Native | Chymotrypsin treated |
|---|---|---|---|
| **Data collection** | 7000 eV | 12398 eV | 12398 eV |
| Space group | I 2 2 2 | I 2 2 2 | P 21 21 21 |
| Cell dimensions | | | |
| $a, b, c$ (Å) | 99.2, 111.7, 119.2 | 99.1, 111.7, 119.3 | 87.2, 120.5, 122.8, |
| $\alpha, \beta, \gamma$ (°) | 90, 90, 90 | 90, 90, 90 | 90, 90, 90 |
| Resolution (Å) | 38.1–2.1 (2.2–2.1)[a] | 31.8–1.8 (1.81–1.75) | 38.9–1.75 (1.78–1.75) |
| $R_{merge}$[b] | 0.10 (1.72) | 0.05 (0.81) | 0.12 (2.69) |
| $I / \sigma I$ | 57.8 (7.8) | 17.4 (2.4) | 23.2 (2.3) |
| Completeness (%) | 97.23 (88.98) | 99.03 (98.74) | 99.1 (85.9) |
| Redundancy | 139.6 (130.1) | 7.5 (7.1) | 37.1 (29.3) |
| **Refinement** | | | |
| Resolution (Å) | 36.9–2.1 (2.2–2.1) | 36.9–1.75 (1.81–1.75) | 38.8–1.75 (1.81–1.75) |
| Total reflections | 4976599 (339216) | 66726 (6289) | 259362 (24593) |
| Unique reflections | 36607 (3616) | 33948 (3301) | 129694 (12300) |
| $R_{work} / R_{free}$[c] | 18.5/20.7 (27.6/29.7) | 17.8/20.9 (33.4/37.0) | 16.4/18.8 (37.0/40.6) |
| Ramachandran favored (%) | 98.2 | 98.4 | 99.4 |
| Ramachandran outliers (%) | 0 | 0 | 0 |
| No. atoms | | | |
| Protein | 3802 | 3854 | 6634 |
| Ligands | 55 | 45 | 73 |
| Water | 288 | 473 | 1130 |
| $B$-factors (Å²) | | | |
| Protein | 45.7 | 27.6 | 31.5 |
| Ligands | 52.8 | 53.3 | 56.6 |
| Water | 46.0 | 44.2 | 42.6 |
| r.m.s. deviations | | | |
| Bond lengths (Å) | 0.002 | 0.011 | 0.014 |
| Bond angles (°) | 0.56 | 1.10 | 1.52 |
| PDB identifier | n/a | 9CYS | 7UBZ |

[a]Values within parentheses refer to the highest resolution shell.
[b]$R_{merge} = \sum\sum_i |I_{hkl} - I_{hkl}(j)| / \sum I_{hkl}$, where $I_{hkl}(j)$ is observed intensity and $I_{hkl}$ is the final average value of intensity.
[c]$R_{work} = \sum ||F_{obs}| - |F_{calc}|| / \sum |F_{obs}|$ and $R_{free} = \sum ||F_{obs}| - |F_{calc}|| / \sum |F_{obs}|$, where all reflections belong to a test set of 10% data randomly selected in Phenix.

synthesized with Tli, and this modification appears necessary for enzymatic activity.

**Structure of the lipase•immunity protein complex**

To identify the Tle modification, we solved the crystal structure of the His$_6$-lipase•Δss-Tli complex to 1.75 Å resolution using sulfur single-wavelength anomalous diffraction (Table 1, S-SAD). The lipase domain adopts an α/β-hydrolase fold with a central 7-strand mixed β-sheet surrounded by 13 α-helices (Fig. 2a). Residues Ser341, Glu393 and His448 (numbered according to full-length Tle) comprise the lipase catalytic triad, with Ser341 projecting from a classical nucleophilic 'elbow' that connects β4 to α7 (Fig. 2a). A DALI server[13] search for structural homologs of the lipase domain recovered plant phospholipase A1 enzymes and several triacylglycerol lipases from fungi (Supplementary Table 1). Δss-Tli is entirely α-helical with six ankyrin repeats

that form an open-solenoidal arc (Fig. 2a, b). The lipase-immunity protein interface is extensive (~3,900 Å²), corresponding to 14.9% of lipase domain and 17.1% of Δss-Tli surface area. The inner helical layer (α8, α10 and α12) of the immunity protein solenoid accounts for most of the contacts with the lipase, though the long loop connecting the 2nd and 3rd ankyrin repeats also forms several hydrogen bonds (Fig. 2b). Overall, Δss-Tli resembles a hand cupped around the lipase, with the ankyrin repeats representing the palm and the connecting loops forming extended fingers. Strikingly, lipase helix α6 is pulled away from the main body of the domain through interactions with Δss-Tli (Fig. 2a, b). AlphaFold2[14] modeling predicts that α6 covers the active site in the absence of Tli (Fig. 2c), suggesting that this helix corresponds to the "lid" found in other lipases[15]. Conformational changes in the lid underlie the phenomenon of interfacial activation, in which lipases act only on phase-separated substrates. The mechanistic basis of interfacial activation is understood for the secreted lipase of *Rhizomucor miehei*, which is a structural homolog of the Tle lipase domain (Fig. 2d, Supplementary Table 1). In aqueous solution, the *R. miehei* lid helix occludes the lipase active site (Fig. 2d)[16]. The active site becomes accessible once the lid is displaced by interfacial tension forces at the aqueous-lipid boundary (Fig. 2d)[17]. The everted helix exposes its hydrophobic face adjacent to the catalytic cleft (Fig. 2e), thereby facilitating penetration into lipid micelles. Surprisingly, Tle appears to adopt an activated conformation when bound to Tli, because the lipase catalytic triad is exposed to solvent in the crystal structure (Fig. 2a, c–e). However, the α6 lid helix of Tle is dislocated well beyond the open position observed in activated *R. miehei* lipase (Fig. 2d). Moreover, the hydrophobic face of α6 packs against the immunity protein (Fig. 2b, d), which should prevent its interaction with lipid substrates. Thus, the immunity protein appears to block lipase activity by sequestering the amphipathic lid helix.

The crystal structure also reveals an unusual interaction between the positively charged sidechains of lipase residues Arg180 from helix α1 and Lys461 in helix α13 (Fig. 2a). F$_o$-F$_c$ difference maps show continuous electron density between these sidechains, suggesting that the residues are covalently linked (Fig. 2f). Arg-Lys crosslinks are known to form upon prolonged exposure to glyoxal or methylglyoxal[18]. These dicarbonyl compounds accumulate as toxic byproducts of metabolism, and they react non-enzymatically with proteins, lipids and nucleic acids to generate a diverse array of advanced glycation end-products (AGEs)[19,20]. AGE protein damage is implicated in the pathology of several chronic degenerative diseases including Alzheimer's, atherosclerosis and diabetes mellitus[21–26]. The +36 Da mass shift observed in the modified lipase domain is consistent with a methylglyoxal-derived imidazolium crosslink (MODIC) (Fig. 1c), and MODIC fits well into the electron density bridging Arg180 and Lys461 (Fig. 2f). MODIC density is also apparent in the crystal structure of chymotrypsin digested lipase•Δss-Tli complex (Supplementary Fig. 2). In the latter structure, chymotrypsin has removed the entire catalytic core of the lipase domain (His340 - Leu445), yet helix α13 remains tethered to α1 via MODIC (Supplementary Fig. 2). Thus, AGE crosslinking joins the N- and C-termini of the lipase domain into a topological circle. This intramolecular crosslink explains increased gel mobility because the lipase cannot be denatured into a fully extended polypeptide and therefore adopts a more compact hydrodynamic radius during electrophoresis.

**AGE crosslinking is required for phospholipase activity**

To confirm the crosslinking site, we generated Arg180Lys and Lys461Gln substitutions in the lipase domain and showed that these variants remain unmodified when produced with Δss-Tli (Fig. 3a, lanes 3 and 7). Mass spectrometry of purified lipase•immunity protein complexes also indicates that the Arg180Lys and Lys461Gln domains do not react with methylglyoxal in vivo (Fig. 3b, c). Because Glu458 forms a direct hydrogen bond with MODIC (Fig. 2f), we also examined a Glu458Gln substitution and observed significantly reduced

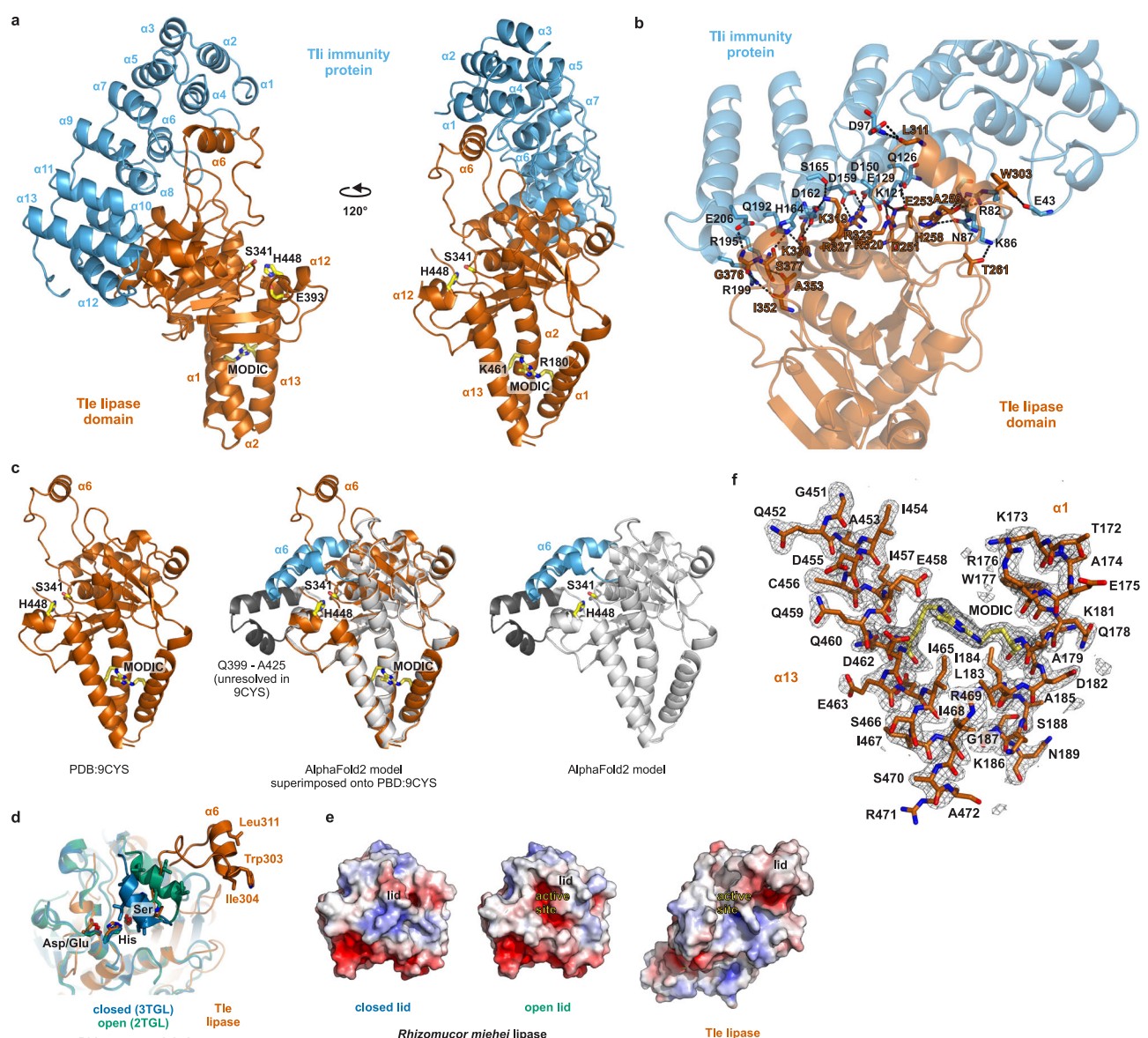

**Fig. 2 | Structure of the lipase•immunity protein complex. a** Structure of the lipase•immunity protein complex. Lipase active-site and MODIC residues are indicated in one-letter code. **b** Hydrogen-bond network of the lipase•Δss-Tli complex interface. **c** The AlphaFold2 model of Tle residues Thr172 - Ala472 superimposed onto the lipase domain structure (PDB: 9CYS). The AlphaFold2 modeled α6 lid helix is shown in cyan, and the unresolved segment from the experimental model is rendered in charcoal. **d** Superimposition of the Tle lipase domain onto inactive (closed) and activated (open) forms of the *R. meihei* tria-cylglycerol lipase. Residues are indicated using three-letter code. **e** Surface electrostatic potentials of lipase domains from panel d. Red surfaces are electronegative and blue surfaces are electropositive. **f** MODIC electron density is illustrated by omit map of lipase helices α1 and α13 at 3 sigma.

crosslinking (Fig. 3a, lane 4). Substitutions of the adjacent Gln459 and Gln460 residues in helix α13 have intermediate effects on MODIC formation (Fig. 3a, lanes 5 and 6). These results demonstrate that Arg180, Glu458 and Lys461 are critical for lipase modification. To determine whether crosslinking is required for enzyme function, we assayed the phospholipase A1 activities of each substituted variant. The Arg180Lys and Lys461Gln lipases have no enzymatic activity, whereas the activities of the other variants are commensurate with the degree of intramolecular crosslinking (Fig. 3a, d). This correlation suggests that MODIC is necessary for lipase function in vitro. We then introduced these mutations into the arabinose-inducible *tle-tli* construct to examine how AGE crosslinking affects Tle-mediated growth inhibition activity. Wild-type *E. cloacae tle*[+] cells inhibit the growth of *E. cloacae* Δ*tle-tli* mutants during coculture, because the latter target cells lack immunity to Tle (Fig. 3e)[12,27]. This competitive advantage is

lost when the *E. cloacae* T6SS is inactivated through a deletion of *tssM* (Fig. 3e, Δ*tssM*), which encodes an essential component of the trans-envelope assembly. Similarly, *E. cloacae* Δ*tle* mutants – which cannot produce Tle effector – are also unable to inhibit Δ*tle-tli* target cells (Fig. 3e). However, *E. cloacae* Δ*tle* cells regain the full competitive advantage when complemented with the wild-type *tle* gene (Fig. 3e). By contrast, plasmids that encode Tle Arg180Lys and Lys461Gln variants fail to restore inhibition activity (Fig. 3e). Immunoblot analysis showed that the Arg180Lys and Lys461Gln variants accumulate at levels comparable to wild-type Tle, but they exhibit no intramolecular cross-linking (Fig. 3f, lanes 4, 5 & 9). The Glu458Gln variant is also significantly attenuated for growth inhibition activity (Fig. 3e) with a corresponding defect in AGE crosslinking (Fig. 3f, lane 6). The Gln459Lys and Gln460Lys lipases retain robust growth inhibition activities (Fig. 3e), which correlate well with in vivo crosslinking

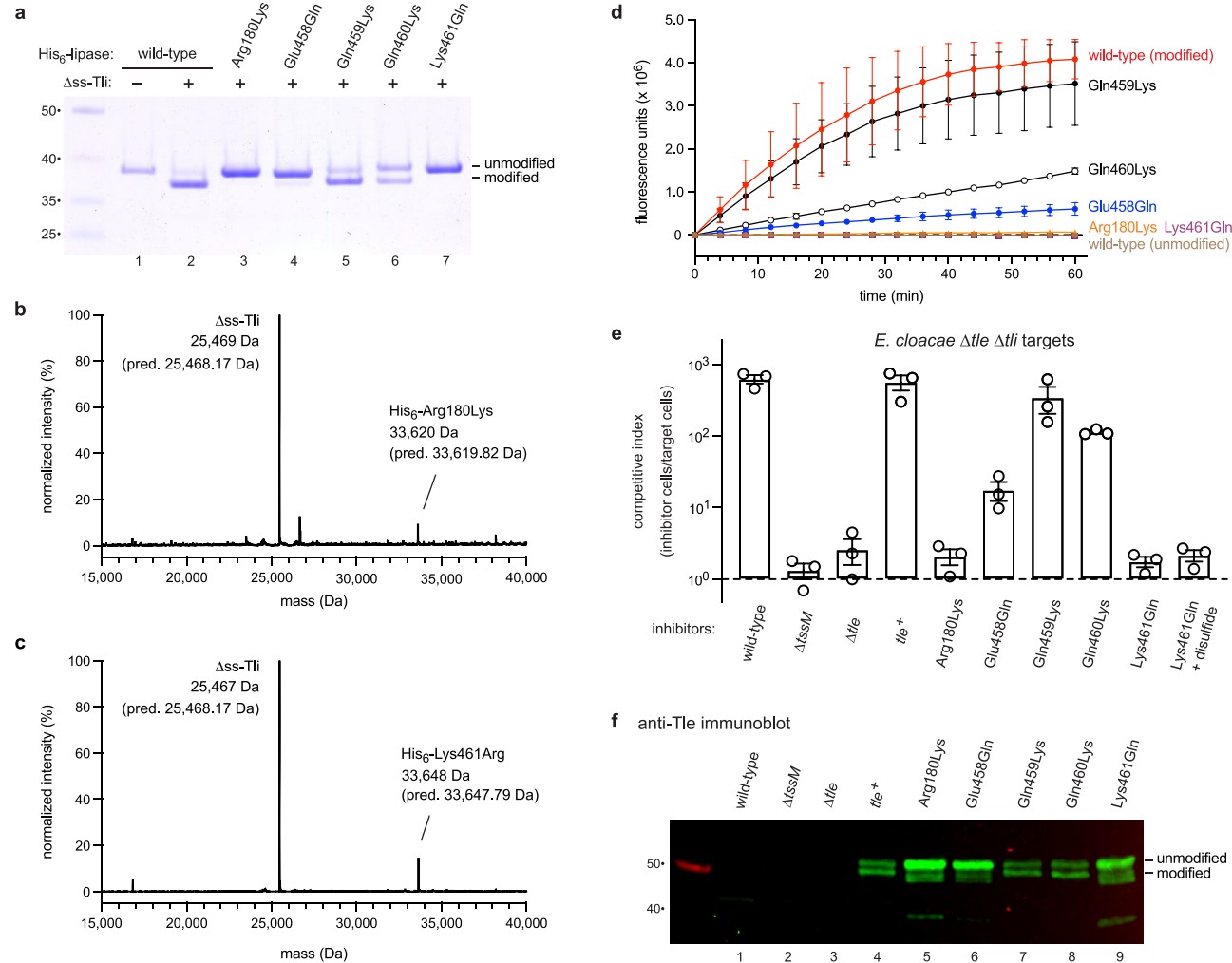

**Fig. 3 | AGE crosslinking is required for phospholipase activity. a** The indicated His₆-lipase domain variants were produced with (or without) Δss-Tli, then purified by Ni²⁺-affinity chromatography under denaturing conditions for SDS-PAGE analysis. Molecular mass markers (kDa) are on the left. **b** Deconvoluted mass spectrum of the His₆-lipase(Arg180Lys)•Δss-Tli complex. **c** Deconvoluted mass spectrum of the His₆-lipase(Lys461Gln)•Δss-Tli complex. **d** In vitro phospholipase A1 activity assays of the purified domains from panel a. Phospholipase activity data are presented as mean values ± standard deviation for three independent experiments.

**e** Competition co-cultures. *E. cloacae* inhibitor strains were co-cultured with Δtle-tli target bacteria and viable cells enumerated as described in Methods. Competitive indices are averages ± standard error for three independent experiments. **f** Urea-soluble cell lysates were prepared from the inhibitor strains in panel e and subjected to immunoblot analysis with polyclonal antibodies to Tle. Molecular mass markers (kDa) are on the left. This experiment was performed independently three times with similar results. Source data are provided as a Source Data file.

efficiency (Fig. 3f, lanes 7 & 8). Collectively, these results indicate that AGE crosslinking is required for the in vitro and in vivo activities of Tle.

**Aldehyde reductases suppress Tle crosslinking**

Endogenous methylglyoxal is produced largely through the non-enzymatic decay of triose phosphates, though enterobacteria encode a methylglyoxal synthase (MgsA) enzyme that generates methylglyoxal during increased flux through the glycolytic pathway[28]. However, MgsA activity does not contribute to crosslinking, because the lipase domain is still modified efficiently when produced in *E. coli* Δ*mgsA* cells (Supplementary Fig. 3, compare lanes 2 and 5). We also attempted to suppress endogenous methylglyoxal levels with aminoguanidine, which reacts with the dicarbonyl to form non-toxic triazine compounds[29]. Treatment with up to 50 mM aminoguanidine has no discernable effect on lipase domain crosslinking in vivo (Supplementary Fig. 3). We next tested whether methylglyoxal detoxification enzymes influence lipase crosslinking. The major detoxification pathway entails condensation with glutathione to form a hemithioacetal, which is subsequently converted into D-lactate by glyoxalase I and II

enzymes[30]. In addition, *E. coli* encodes a glutathione-independent glyoxalase, HchA, and a number of NAD(P)H-dependent aldehyde reductases capable of methylglyoxal detoxification[30]. We cloned various detoxification genes under the control of an arabinose-inducible promoter and asked whether their overexpression affects Tle crosslinking in vivo. Somewhat surprisingly, glyoxalase I (GloA) overexpression has little effect (Fig. 4a, lane 3), but HchA induction suppresses Tle crosslinking (Fig. 4a, lane 4). The YdjG and YqhD-YqhE aldehyde reductases also appear to decrease crosslinking efficiency in *E. coli* cells (Fig. 4a, lanes 5 and 6). We next introduced these plasmids into *E. cloacae* to test whether the detoxification enzymes decrease the level of activated Tle available for T6SS export. Each construct reduces the competitive fitness of wild-type *E. cloacae* against Δ*tle-tli* target bacteria, though the effects are modest (Fig. 4b). HchA has the greatest effect with a ~10-fold decrease in the competitive index (Fig. 4b). We note that these detoxification enzymes have Michaelis constants (Kₘ) for methylglyoxal in the low millimolar range[31–34], suggesting that the dicarbonyl is still present at micromolar concentrations in these cells.

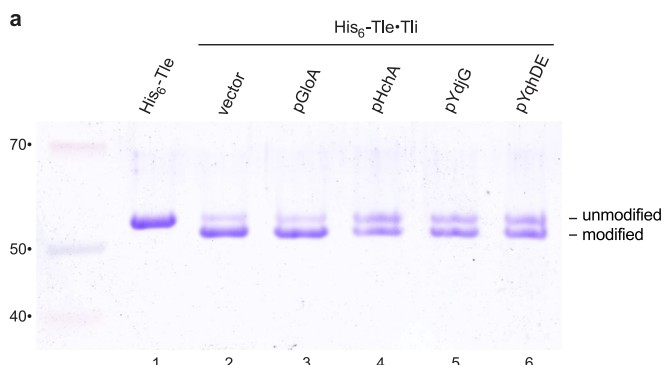

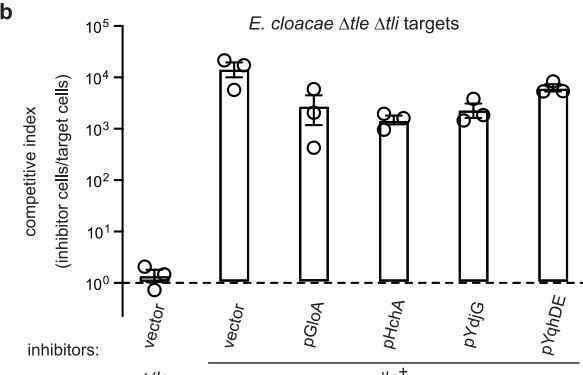

**Fig. 4 | Aldehyde reductases suppress Tle crosslinking. a** His$_6$-Tle was produced with Tli in *E. coli* cells that overexpress the indicated methylglyoxal detoxification enzymes. His$_6$-Tle was purified by Ni$^{2+}$-affinity chromatography under denaturing conditions and analyzed by SDS-PAGE. Molecular mass markers (kDa) are on the left. **b** Competition co-cultures. *E. cloacae* inhibitor strains that overexpress methylglyoxal detoxification enzymes were co-cultured with *E. cloacae* Δtle-tli target bacteria and viable cells enumerated as described in Methods. Competitive indices are averages ± standard error for three independent experiments. Source data are provided as a Source Data file.

### In vitro activation of the Tle phospholipase domain

Given that AGE crosslinking is not enzyme-catalyzed, it should be possible to activate the lipase domain by methylglyoxal treatment in vitro. We mixed unmodified His$_6$-lipase with Δss-Tli and treated the purified complex with 1 mM methylglyoxal to monitor crosslink formation as a function of time. Intramolecular crosslinking is detectable after 1 h and increases up to the 24 h endpoint (Supplementary Fig. 4). We also examined different methylglyoxal concentrations and found that crosslinking is apparent at 100 μM methylglyoxal and nears completion at 5 mM methylglyoxal (Fig. 5a). The lipase can also be crosslinked with glyoxal at lower efficiency (Fig. 5b). As predicted from the mutagenesis data presented above, the Arg180Lys and Lys461Gln lipase variants are not subject to methylglyoxal crosslinking in vitro (Fig. 5c, d). Moreover, lipase crosslinking is not observed in the absence of Δss-Tli (Supplementary Fig. 5a), indicating that the domain only reacts with methylglyoxal when bound to immunity protein. To determine whether the in vitro crosslinked lipases are active, we first isolated the His$_6$-lipase domains away from Δss-Tli by Ni$^{2+}$-affinity chromatography under denaturing conditions. After refolding by dialysis, the activities of dicarbonyl treated lipases were determined. Remarkably, glyoxal and methylglyoxal crosslinking both promote phospholipase A1 activity, whereas the untreated lipase remains inert (Fig. 5e). Somewhat surprisingly, glyoxal treatment leads to greater enzymatic activity (Fig. 5e), despite more efficient crosslinking with methylglyoxal (Fig. 5a, b). However, we subsequently found that the glyoxal crosslink is thermolabile (Supplementary Fig. 5b, lanes 5 and 6), and therefore glyoxal crosslinking is underestimated in Fig. 5b because these samples were heated to 95 °C prior to SDS-PAGE analysis. Importantly, the in vitro crosslinked lipases can be neutralized with purified Δss-Tli (Fig. 5e), indicating that the treated domains retain native interactions with the immunity protein.

### Phospholipase activation with an engineered disulfide bond

AGE crosslinking presumably promotes lipase function by stabilizing tertiary structure. Thermodynamic stability is particularly important for lipases, which necessarily operate at the water-lipid interface. Interfacial tension at this phase boundary is known to disrupt protein structure and inactivate enzymes[35,36]. Thus, the same interfacial forces that promote lipase activity through lid displacement also have the potential to inactivate the enzyme[37]. We note that the lipase of *R. meihei* contains a conserved Cys123-Cys362 disulfide bond that tethers the N- and C-termini (Fig. 6a). Given that disulfides protect proteins from denaturation at the water-lipid interface[36], we reasoned that this bond may serve the same stabilizing function as MODIC in Tle. If this

hypothesis is correct, then an engineered disulfide bond should be able to restore phospholipase activity to the Lys461Gln domain. Disulfide-by-design[38] software predicts that Cys substitutions of Tle residues Lys186 and Ala472 should form a disulfide linking helices α1 and α13 (Fig. 6b). Indeed, the resulting lipase variant forms an intramolecular disulfide bond under oxidizing conditions when produced with Δss-Tli (Fig. 6c, lanes 5 & 6). In contrast to the Lys461Gln domain, the disulfide-linked variant has significant phospholipase A1 activity that is neutralizable with the Δss-Tli immunity protein (Fig. 6d). Moreover, phospholipase activity is diminished when the disulfide bond is reduced with dithiothreitol (Fig. 6d). The residual activity in the latter experiment presumably reflects lipase that is resistant to reduction (Supplementary Fig. 6a, lanes 3 and 4). Importantly, dithiothreitol has no effect on the activity of MODIC modified wild-type lipase (Fig. 6d). Together, these data demonstrate that intramolecular crosslinking is necessary and sufficient for Tle phospholipase activity. Although the disulfide-linked lipase domain is active in vitro, the corresponding full-length Tle variant does not inhibit target bacteria in competition cocultures (Fig. 3e). Immunoblotting shows that the effector does not form an intramolecular disulfide in vivo (Supplementary Fig. 6b, lanes 3 and 4), consistent with the lack of growth inhibition activity in bacterial competitions.

### Discussion

*E. cloacae* cells produce two forms of Tli immunity protein with distinct functions. Full-length Tli is secreted to the periplasm, where it is lipidated and embedded into membranes to protect against incoming Tle effectors (Fig. 7). Cytosolic Tli is synthesized from alternative translation start codons found downstream of the signal sequence[12]. The latter pool of immunity protein is required to activate Tle by promoting methylglyoxal crosslinking, suggesting that cytosolic Tli forms a transient complex with newly synthesized Tle prior to T6SS-mediated export (Fig. 7). MODIC formation must be efficient under physiological conditions, because T6SS effectors are deployed within minutes of synthesis[3]. Moreover, endogenous methylglyoxal concentrations are estimated to be in the low micromolar range[39], indicating that the Tle•Tli complex has high affinity for the dicarbonyl. The mechanism by which Tli promotes AGE crosslinking is not clear. The immunity protein does not make contact with helix α1 or α13 of the lipase domain, and the MODIC residue is ~33 Å from Tli in the crystal structure. These observations suggest that Tli does not directly catalyze AGE crosslinking. Given that unmodified Tle is insoluble, perhaps Tli acts as a folding chaperone to guide the lipase into a conformation that is competent for crosslinking. This model implies an additional mechanism to dissociate

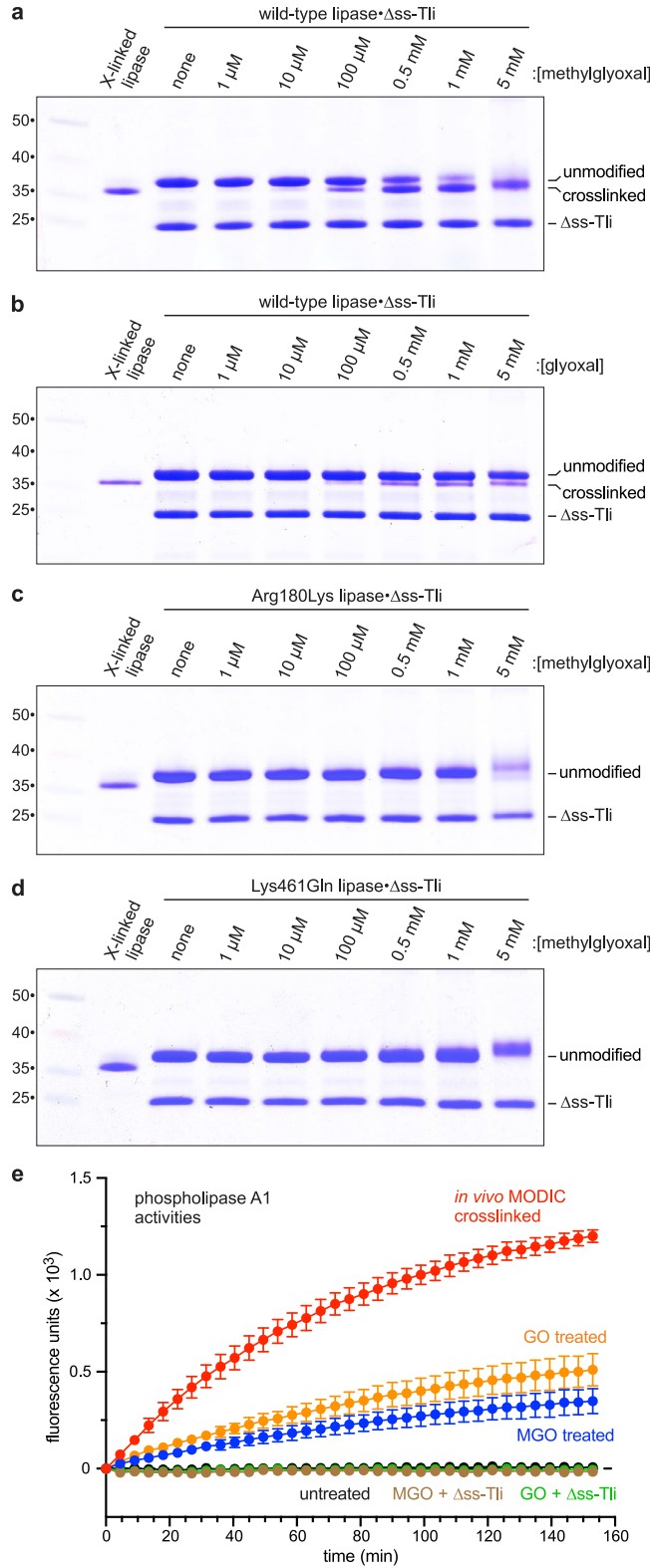

**Fig. 5 | In vitro activation of the phospholipase domain.** Unmodified lipase domain and Δss-Tli were mixed in vitro to form a complex, which was then treated with methylglyoxal (**a**) or glyoxal (**b**) at the indicated concentrations for 7 h at 37 °C. Treated complexes were analyzed by SDS-PAGE. Methylglyoxal treatment of Arg180Lys (**c**) and Lys461Gln (**d**) lipase domains in complex with Δss-Tli. Molecular mass markers (kDa) are on the left. **e** In vitro crosslinked His$_6$-lipase domains were isolated from Δss-Tli by Ni$^{2+}$-affinity chromatography under denaturing conditions, then refolded by dialysis for phospholipase A1 activity assays. Data are presented as mean values ± standard deviation for three independent experiments. Source data are provided as a Source Data file.

and fatty acids as nutrients. Before secretion, the newly synthesized lipases are first transferred into the endoplasmic reticulum lumen, where disulfide bond formation is orchestrated through the redox activities of Ero1p and Pdi1p[42]. The lipases are then trafficked through the Golgi apparatus and ultimately secreted via exocytosis[43]. By contrast, T6SS effectors are transferred directly from the inhibitor-cell cytoplasm to the periplasm of target bacteria (Fig. 7). Because the bacterial cytosol is a reducing environment[44], T6SS effectors presumably cannot form disulfides prior to export, which may explain why the engineered disulfide approach fails to activate Tle in competition cocultures. Coulthurst and coworkers have shown that some T6SS effectors can acquire disulfide bonds after delivery into the oxidizing environment of the target-cell periplasm[45]. However, like the MODIC residue, the engineered disulfide in Tle only forms when the effector is bound to immunity protein. These constraints presumably underlie the evolutionary pressure for Tle to utilize an AGE crosslink in lieu of a disulfide bond.

Mougous and coworkers originally delineated five families of T6SS phospholipase effectors (Tle1-Tle5)[46]. *E. cloacae* Tle does not belong to any of these families, though it is homologous to the Slp lipase effector from *Serratia marsescens*[47]. We propose that this new phospholipase effector group be designated Tle6 in accordance with the nomenclature of Russel et al. [46]. Tle6 effectors are invariably encoded with ankyrin-repeat containing immunity proteins[12], raising the possibility that these lipases are all subject to Tli-dependent AGE crosslinking. By contrast, representative Tle1, Tle2, Tle4 and Tle5 lipases are active in vitro when purified from heterologous expression systems[46,48,49]. Moreover, Tle1, Tle2 and Tle5 are readily deployed from mutant strains that lack the corresponding immunity genes[46,48,50]. Given that other T6SS lipase effectors are intrinsically active, the advantage of AGE-mediated Tle6 activation is unclear. This phenomenon could represent a fail-safe mechanism that prevents auto-intoxication in Tle6 producing cells. The inner leaflet of the cytoplasmic membrane is composed of glycerophospholipid, and therefore cytosolic phospholipases could disrupt cell integrity. Activation in the context of the Tle6-Tli6 complex would ensure that the phospholipase remains neutralized until export (Fig. 7). Auto-intoxication is potentially a problem for other T6SS phospholipases as well, though these enzymes are generally not active when produced in the cytosol[46,48,49,51]. In some instances, Tle effectors have been shown to be toxic only when directed to the periplasm with signal peptides[46,48,49,52]. The mechanistic basis of this activation has not been explored, but the redox potential and increased Ca$^{2+}$ ion content of the periplasm could conceivably modulate phospholipase activity.

In summary, decades of research have concluded that glycation adducts are invariably deleterious to protein function[23,53]. However, our results show that methylglyoxal-mediated AGE crosslinking can be harnessed as a bona fide post-translational modification to promote protein function. Given that reactive dicarbonyl compounds accumulate in all cells – prokaryotic and eukaryotic – these findings raise the possibility that AGE crosslinking could be exploited to enhance the function of other proteins. This alternative strategy to fortify protein structure could be particularly advantageous in the cytoplasm, where the redox potential precludes disulfide bond formation.

Tli before the activated effector is loaded into the T6SS (Fig. 7). In principle, binding interactions between Tle and Hcp or VgrG could provide the free energy to remove Tli during packaging.

We propose that AGE crosslinking activates Tle by stabilizing its tertiary structure to resist denaturation at water-lipid phase boundaries. Structurally related triacylglycerol lipases from fungi use long-range disulfide bonds to address this biophysical problem[16,40,41]. These neutral lipases are released into the environment to scavenge glycerol

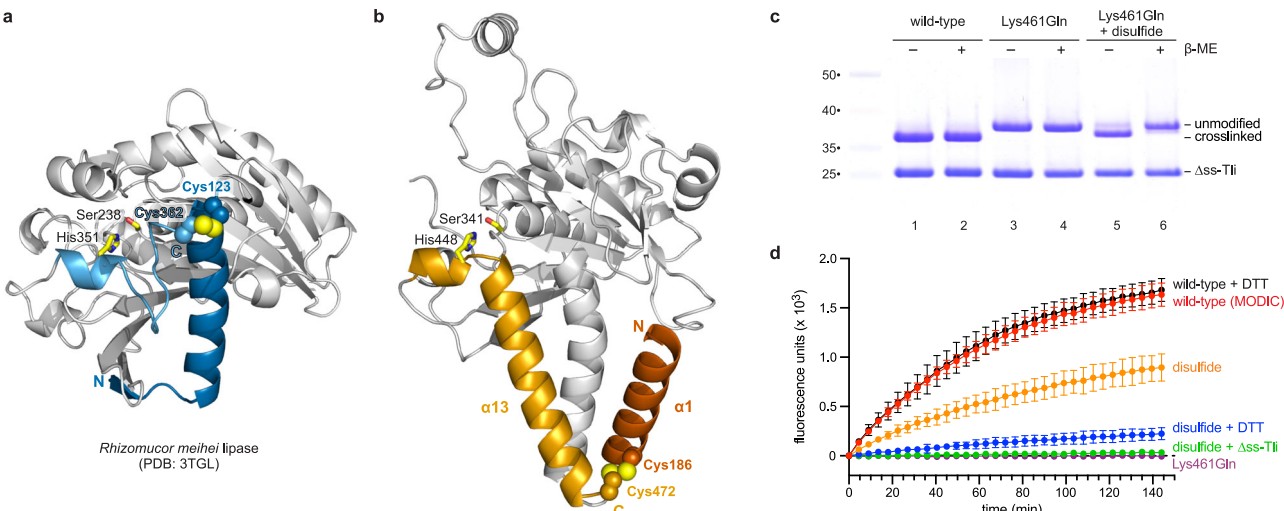

**Fig. 6 | Phospholipase activation with an engineered disulfide bond. a** Structure of *R. meihei* lipase illustrating the disulfide bond linking the N- and C-termini. **b** Predicted structure of Disulfide-by-design engineered Tle phospholipase domain. **c** The indicated His$_6$-lipase variants were produced with Δss-Tli in *E. coli* cells, and the complexes purified by Ni$^{2+}$-affinity chromatography under non-denaturing conditions. Complexes were analyzed by SDS-PAGE under oxidizing and reducing (β-mercaptoethanol, β-ME) conditions. Molecular mass markers

(kDa) are on the left. **d** The indicated His$_6$-lipase domains were isolated from Δss-Tli by Ni$^{2+}$-affinity chromatography under denaturing conditions, then refolded by dialysis for phospholipase A1 activity assays. Phospholipase activity data are presented as mean values ± standard deviation for three independent experiments. Reactions were supplemented with dithiothreitol (DTT) where indicated. Source data are provided as a Source Data file.

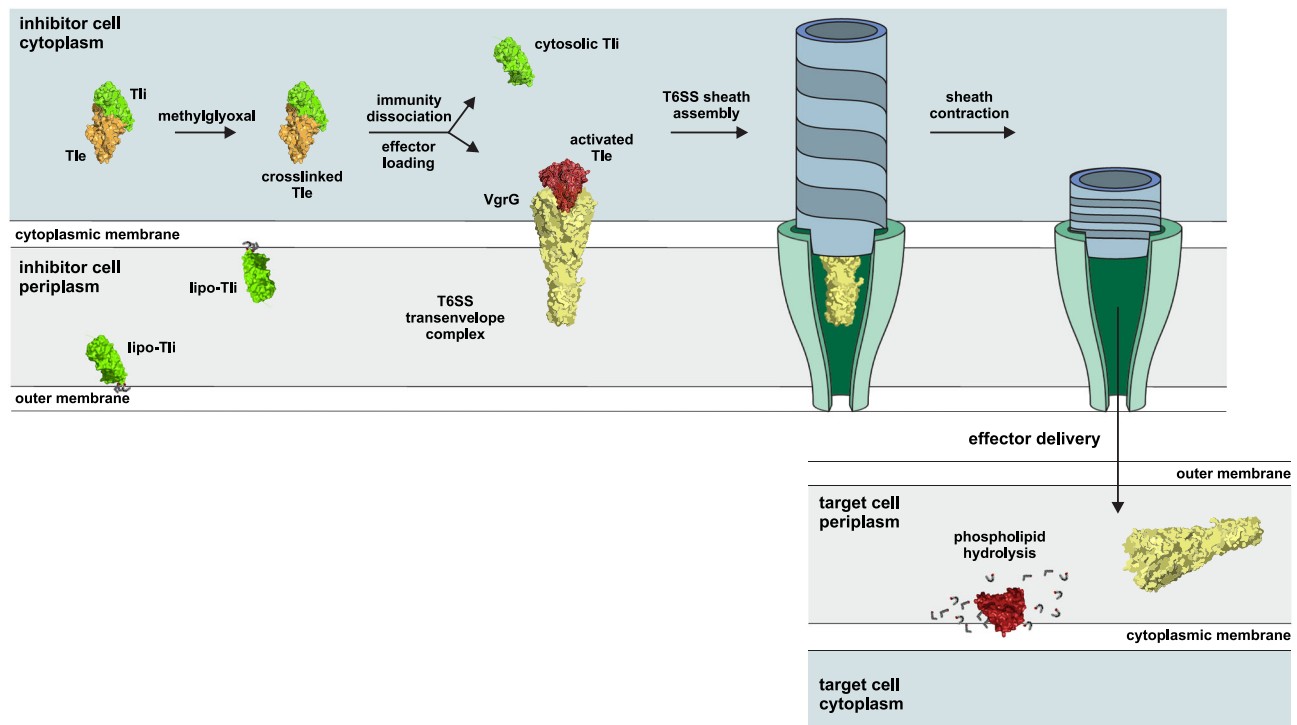

**Fig. 7 | Model for Tli-dependent Tle activation and T6SS mediated delivery.** Mature Tli (lipo-Tli) is secreted to the periplasm where it is lipidated and embedded in membranes to protect the cell from incoming Tle effectors. *E. cloacae* cells also produce cytosolic Tli, which presumably forms a transient complex with Tle to enable methylglyoxal crosslinking. The activated effector is then dissociated from the immunity protein and loaded into the T6SS apparatus for export. Upon delivery into the target-cell periplasm, activated Tle hydrolyzes membrane phospholipids.

## Methods

### Bacterial strain and plasmid constructions

Bacterial strains and plasmids are listed in Supplementary Table 2. The Δ*mgsA::kan* allele from the Keio strain collection[54] was transferred into strain CH6247 by phage P1 mediated transduction to generate strain CH5691. Oligonucleotide primers used for plasmid constructions are presented in Supplementary Table 3. All bacteria were grown at 37 °C in lysogeny broth (LB): 1% tryptone, 1% yeast extract, 0.5% NaCl. Media were supplemented with antibiotics at the following concentrations: ampicillin (Amp), 150 μg/mL; kanamycin (Kan), 50 μg/mL; rifampicin

(Rif), 200 µg/mL; spectinomycin (Spc), 100 µg/mL; tetracycline (Tet), 15 µg/mL; and trimethoprim (Tp), 100 µg/mL, where indicated.

The Δss-Tli coding sequence was amplified with primers CH3719/CH3419 and ligated to pET21K and pCH450K using KpnI/XhoI restriction sites to generate plasmids pCH3291 and pCH495, respectively. The *tli* gene was also amplified with CH5751/CH3419 and ligated to pMCSG63 via KpnI/XhoI sites to generate plasmid pCH7695, which appends a tobacco etch virus (TEV) protease cleavable His$_6$ epitope to the N-terminus of Δss-Tli. Full-length *tle* was amplified with CH4703/ZR248 and the product ligated to pCH8259[55] and pSH21[56] using SpeI/XhoI to produce plasmids pCH7620 and pCH2199, respectively. Plasmid pCH5892 was generated by subcloning the *tli* containing HindIII/XhoI fragment from pCH3128 into pCH2199. The coding sequence for the Tle lipase domain was amplified with CH5807/ZR248 and ligated to pCH8259 to generate plasmid pCH7088. Fragments containing *tle* and *tle-tli* were subcloned from pCH14212 and pCH3128[12] into pSCBAD[57] using BamHI/XhoI restriction sites to generate plasmids pCH2826 and pCH4362, respectively. Missense mutations were introduced into *tle* using forward primers CH5721 (Arg180Lys), CH5689 (Glu458Gln), CH5722 (Gln459Lys), CH5723 (Gln460Glu), CH5690 (Lys461Gln) in conjunction with CH3419. The resulting products were used as megaprimers in conjunction with CH4469 to amplify *tle-tli*, and the final products ligated to pCH4362 using KpnI/XhoI restriction sites to generate plasmids pCH5501 (Arg180Lys), pCH7600 (Glu458Gln), pCH5502 (Gln459Lys), pCH5503 (Gln460Glu) and pCH7601(Lys461Gln). The disulfide lipase variant containing Lys186Cys and Ala472Cys substitutions was generated by three rounds of overlap extension PCR using plasmid pCH4362 as a template. The endogenous Cys456 residue was first mutated to Ser in the *tle(K461Q)* allele with primers CH943/CH5960 and CH5959/CH3419. The Ala472Cys (using primers CH943/CH6063 and CH6062/CH3419) and Lys186Cys (CH943/CH5908 and CH5907/CH3419) mutations were then introduced sequentially. The final product was digested with BamHI/XhoI and ligated to pSCBAD to generate plasmid pCH3923.

His$_6$-tagged lipase domain constructs were amplified using primers CH5087/ZR248 and ligated to pCH8259 using SpeI/XhoI restriction sites to generate plasmids pCH8871 (Glu458Gln), pCH8717 (Gln459Lys), pCH8718 (Gln460Glu) and pCH8872 (Lys461Gln). The catalytically inactive *tle(S321A)* allele was amplified from pCH3763[12] with primers CH5087/ZR248 and ligated to pCH8259 using SpeI/XhoI site to generate plasmid pCH3936. The *tle(R180K)* allele was amplified with CH5688/ZR248 and ligated to pCH8259 using SpeI/XhoI sites to generate plasmid pCH7599. The *tle(K186C,C456S,K461Q,A472C)* allele was amplified with CH5087/CH5909 and ligated to pCH8259 using SpeI/XhoI sites to generate plasmid pCH1382.

Methylglyoxal detoxification genes were amplified from *E. coli* genomic DNA and placed under the control of an arabinose-inducible promoter on plasmid pSCBAD. *gloA* was amplified with primers CH5761/CH5762 and ligated using EcoRI/XhoI restriction sites. *hchA* was amplified with CH6310/CH6311 and the product digested with MfeI/SbfI for ligation to EcoRI/PstI-digested pSCBAD. *ydjG* was amplified with CH6312/CH6313 and ligated using EcoRI/PstI sites. The *yqhD-yqhE* gene pair was amplified with CH6314/CH6315 and ligated via EcoRI/PstI sites.

## Tle-Tli coproduction

*E. coli* X90 and *E. cloacae* ATCC 13047 cells were transformed with pCH2826 (*tle*) or pCH4362 (*tle-tli*), and cells were grown in Tp-supplemented LB media at 37 °C. Once the cultures reached an optical density at 600 nm (OD$_{600}$) of ~ 0.4, they were split in two and one half was induced with 0.4% L-arabinose. After 1 h, cells were collected by centrifugation at 21,130*g* in a microfuge and the cell pellets frozen at −80 °C. Frozen pellets were resuspended in urea lysis buffer [8 M urea, 20 mM Tris-HCl (pH 8.0), 100 mM NaCl] and subjected to another freeze-thaw cycle to break cells. Lysates were clarified by

centrifugation at 21,130*g* in a microfuge. Protein concentrations were estimated using Bradford reagent and equal amounts of lysate were analyzed by SDS-PAGE at 110 V using 7% polyacrylamide gels buffered with Tris-tricine. Gels were soaked for 10 min in 25 mM Tris, 192 mM glycine (pH 8.6), 20% methanol before transfer to polyvinylidene fluoride membranes at 17 V for 30 min using a semi-dry electroblotting apparatus. Membranes were blocked with 4% non-fat milk in 1× PBS for 45 min at ambient temperature, then incubated with a 1:5,000 dilution of rabbit anti-Tle polyclonal antisera[12] in 1× PBS, 4% non-fat milk overnight. After three 10 min washes with 1× PBS, the membranes were incubated with IRDye 800CW-conjugated goat anti-rabbit IgG (LI-COR, 1:40,000 dilution) in 1× PBS for 45 min. Immunoblots were visualized using a LI-COR Odyssey infrared imager. The effect of methyglyoxal detoxification on AGE crosslinking was assessed in *E. coli* cells that overproduce His$_6$-Tle and Tli from an IPTG-inducible phage T7 promoter. *E. coli* CH2016 cells carrying plasmid pCH5892 (*his$_6$-tle/tli*) were transformed with pCH7694 (*gloA*), pCH5119 (*hchA*), pCH5120 (*ydjG*) or pCH5121(*yqhDE*). The resulting strains were grown in LB media supplemented with Amp Tp to mid-log phase, and detoxifying enzyme expression was induced with 0.4% L-arabinose for 90 min. His$_6$-Tle/Tli production was then induced with 1.5 mM IPTG for 30 min. Cells were collected by centrifugation, and His$_6$-Tle was purified by Ni$^{2+}$-affinity chromatography under denaturing conditions as described above. Purified proteins were resolved by SDS-PAGE at 110 V on 7% polyacrylamide gels buffered with Tris-tricine.

## His$_6$-lipase domain modification and purification

To examine lipase domain modification, *E. coli* CH2016 cells[58] were co-transformed with pCH7088 (*his$_6$-lipase*) or pCH3936 (*his$_6$-lipase-S341A*) and pET21b or pCH3291 (Δss-tli). The resulting strains were grown to OD$_{600}$ ~ 1.0 in LB media supplemented with Amp and Tet, and lipase domain production was induced with 0.4% L-arabinose for 1 h. Where indicated, aminoguanidine was added to cultures at the same time as L-arabinose. Δss-Tli expression was then induced with 1.5 mM IPTG for 90 min. Cell pellets were collected by centrifugation at 4303*g* for 10 min and frozen at −80 °C. To isolate His$_6$-lipase•Δss-Tli complexes, cells were resuspended in native lysis buffer [20 mM Tris-HCl (pH 7.5), 150 mM NaCl, 0.05% Triton X-100, 5 mM imidazole] and adjusted to 0.1 mg/mL lysozyme for 15 min prior to sonication. Sonicated cell lysates were clarified by centrifugation at 14,462*g* for 10 min at 4 °C, and the supernatant incubated with Ni$^{2+}$-nitrilotriacetic acid (Ni$^{2+}$-NTA) agarose for 90 min at ambient temperature. Ni$^{2+}$-NTA resins were washed five times with 10 volumes of native lysis buffer, and His$_6$-lipase•Δss-Tli complexes were eluted in native elution buffer [20 mM Tris-HCl (pH 7.5), 150 mM NaCl, 0.05% Triton X-100, 250 mM imidazole]. To isolate His$_6$-lipase domains, cell pellets were resuspended in guanidine lysis buffer [6 M guanidine-HCl, 20 mM Tris-HCl (pH 8.0), 1 mM imidazole] and sonicated. Lysates were clarified by centrifugation at 14,462*g* for 10 min at 4 °C, and the supernatants incubated with Ni$^{2+}$-NTA agarose for 90 min at ambient temperature. Resins were washed four times with 10 volumes of 8 M urea, 1 mM imidazole, and His$_6$-lipase domains eluted with 8 M urea, 20 mM Tris-HCl (pH 8.0), 250 mM imidazole. Purified lipase domains were refolded by dialysis against 20 mM sodium phosphate (pH 7.0). His$_6$-lipase domain concentrations were quantified by absorbance at 280 nm (A$_{280}$) using an extinction coefficient of 35,410 M$^{-1}$ cm$^{-1}$. Modified full-length His$_6$-Tle was purified in the same manner under denaturing conditions and quantified using an extinction coefficient of 66,810 M$^{-1}$ cm$^{-1}$.

## Δss-Tli purification

*E. coli* CH2016 cells carrying plasmid pCH7695 were grown in Amp-supplemented LB medium at 37 °C. At OD$_{600}$ ~ 1.0, His$_6$-TEV-Δss-Tli production induced with 1.5 mM IPTG for 90 min. Cell were collected by centrifugation at 4303*g* for 10 min and frozen at −80 °C. Frozen cell pellets were resuspended in guanidine lysis buffer, and His$_6$-TEV-Δss-

Tli was purified as described above for His$_6$-lipases. After dialysis into 20 mM sodium phosphate (pH 7.0), the N-terminal His$_6$ epitope was removed by digestion with tobacco etch virus (TEV) protease overnight at 37 °C in 20 mM sodium phosphate (pH 7.0) supplemented with 2 mM dithiothreitol (DTT). His$_6$-TEV protease and released His$_6$ epitopes were removed by incubation with Ni$^{2+}$-NTA agarose for 90 min. Untagged Δss-Tli was quantified by A$_{280}$ using an extinction coefficient of 27,960 M$^{-1}$ cm$^{-1}$.

## Phospholipase activity assays

Tle and lipase domain activities were measured by monitoring the hydrolysis of fluorogenic phospholipase substrates PED-A1 (PLA$_1$-specific) and PED6 (PLA$_2$-specific). PED-A1 and PED6 (1.5 μM) were incorporated into liposomes of dioleoylphosphatidylcholine (DOPC, 15 μM) and dioleoylphosphatidylglycerol (DOPG, 15 μM) according manufacturer's instructions (Invitrogen). His$_6$-Tle/His$_6$-lipase domains were assayed at 600 nM, and immunity blocking experiments were conducted by supplementing the reactions with 700 nm purified Δss-Tli. Assays were performed in 50 mM Tris-HCl (pH 7.5), 100 mM NaCl, 1 mM CaCl$_2$. Fluorescence was measured every 4 min for 1 h using a PerkinElmer Victor 3 V Multilabel Plate Reader (Figs. 1d, e, 3d, Supplementary . 1c, d), or every 4.5 min for 153 min using a Molecular Devices SpectraMax M5 fluorimeter (Fig. 5e & 6d). Fluorescence values were baseline corrected by subtracting the background fluorescence from mock control reactions containing only buffer and fluorogenic substrate. All assays were performed independently three times.

## Mass spectrometry

Purified His$_6$-Tle•Δss-Tli and His$_6$-lipase•Δss-Tli complexes (5 μL at 0.02 mg/mL) were analyzed by mass spectrometry in a single technical replicate ($n$ = 1 for each construct analyzed) using LC-MS (ACQUITY UPLC H-class system, Xevo G2-XS QTof, Waters). Samples were loaded onto a phenyl guard column at 45 °C (ACQUITY UPLC BEH Phenyl VanGuard Pre-column, 130 Å, 1.7 μm, 2.1 mm × 5 mm, Waters) in 0.1% formic acid in water at a flow rate of 0.2 mL/min. Proteins were eluted with a linear gradient to 90% acetonitrile over 5 min. The Xevo Z-spray source was run in positive ion mode with a capillary voltage of 300 V, and a cone voltage of 40 V (NaCsI calibration, Leu-enkephalin lockmass), 400–4000 Da. Nitrogen was used as the desolvation gas at 350 °C and a flow rate of 800 L/h. Total average mass spectra were reconstructed from the charge state ion series using the MaxEnt1 algorithm[59] from Waters MassLynx software V4.1 SCN949 according to the manufacturer's instructions. To obtain the ion series described, the major peak of the chromatogram was selected for integration before further analysis.

## X-ray crystallography

*E. coli* CH2016 was transformed with pCH15269 and pCH495[12] to generate a strain that over-produces His$_6$-lipase and Δss-Tli. Cells were grown in LB supplemented with 50 μg/mL Amp and 15 μg/mL Tet at 37 °C to OD$_{600}$ - 0.6-0.7, and immunity protein expression was induced with 0.4% L-arabinose for 1 h. Lipase domain expression was then induced with 0.5 mM IPTG for 30 min. Cells were harvested by centrifugation for 30 min at 3100$g$ and the pellets stored at −20 °C. Cells were broken by sonication in Buffer A (20 mM sodium phosphate (pH 7.0), 150 mM NaCl) with phenylmethylsulfonyl fluoride and lysozyme. Cell debris was removed by centrifugation at 22,000$g$ for 1 h, followed by filtration of the supernatant through a 0.45 μm filter. The complex was purified on a HisTrap column (GE healthcare) in Buffer A. The His$_6$-lipase•Tli complex was eluted with a linear gradient of 500 mM imidazole in Buffer A. Fractions containing the His$_6$-lipase•Tli complex were identified by SDS-PAGE, and concentrated for further purification on a S200/10/300 column (GE healthcare) in 20 mM Tris pH 7.5, 150 mM NaCl. Fractions with pure His$_6$-lipase•Tli complex were identified by SDS-PAGE analysis, concentrated to 80 mg/mL and

screened for crystallization at room temperature and 4 °C. After 6 months, crystals were found in PACT 59 (0.2 M sodium citrate, 20% PEG 3350) at 4 °C. Crystallization was optimized and reproduced in 0.4 M sodium citrate, 22% PEG 3350. Crystals were collected after three weeks in mother liquor supplemented with 20% glycerol and cryo-cooled before data collection at SSRL beamline 12-2. Native (at 12398 eV) and sulfur single-wavelength anomalous diffraction (S-SAD, at 7000 eV) datasets were collected from a single crystal. The S-SAD data was processed by XDS (Versions January 31, 2020; February 5, 2021; January 10, 2022)[60] and scaled and truncated in Aimless (imosflm version 7.4.0) to 2.1 Å resolution[61]. The native dataset was handled identically and truncated to 1.75 Å (Table 1). Phasing and initial model building were performed with the S-SAD datasets in Crank2 in ccp4i2 (Version 1.1.0 revision 6539, user interface to CCP4 Program Suite version 9.0.003)[62]. Subsequent model building and refinement were performed in phenix.refine (PHENIX 1.19.1_4122) and coot (Coot 0.9.3 to 0.9.8.8)[63,64]. The S-SAD dataset was used to solve the native datasets by MR in Phaser (PHENIX 1.19.1_4122)[63], and refined as described above to a final $R_{work}$ and $R_{free}$ of 17.8 and 20.9%.

Chymotrypsin was used as an additive to facilitate crystallization through limited proteolysis during initial trials. Chymotrypsin was added to the His$_6$-lipase•Δss-Tli complex at a final concentration of 0.02 mg/mL. Proteolyzed His$_6$-lipase•Δss-Tli was crystallized in 0.25 M sodium citrate, 22% PEG-3350 and cryo-cooled in mother liquor supplemented with 20% glycerol. A dataset was collected at SSRL beamline 12-2 at 12659 eV. Phasing was initially determined in ARP-WARP (Version 8.0)[65] with a search for Tli. Subsequently, the lipase domain structure was built partially in autobuild (PHENIX 1.19.1_4122)[66]. The remainder of the lipase domain was built and refined manually as described above to a final $R_{work}$ and $R_{free}$ of 16.4% and 18.8%.

## Competition co-cultures

Inhibitor and target cells were grown overnight in LB media, then diluted 1:50 into fresh LB media and grown at 37 °C to mid-log phase. Cells were harvested by centrifugation at 3400$g$ for 2 min, resuspended to an OD$_{600}$ of 3.0 and mixed at a 10:1 ratio of inhibitor to target cells. Cell mixtures (10 μL) were spotted onto LB-agar supplemented with 0.4% L-arabinose and incubated at 37 °C. A portion of each cell mixture was serially diluted into 1× M9 salts and plated onto antibiotic supplemented LB-agar to enumerate inhibitor and target cells as colony forming units (cfu) at $t$ = 0 h. Inhibitor cells were scored by Tp$^R$ (conferred by plasmid pSCBAD derivatives), and target cells were enumerated by Rif$^R$. After 4 h of co-culture, cells were harvested from the agar surface with polyester-tipped swabs and resuspended in 1.0 mL of 1× M9 salts. To examine the effects of methylglyoxal detoxifying enzymes, wild-type *E. cloacae* inhibitor cells harboring plasmids pCH7694 (*gloA*), pCH5119 (*hchA*), pCH5120 (*ydjG*) or pCH5121(*yqhDE*) were co-cultured with *E. cloacae* Δ*tle* Δ*tli* target bacteria at a 1:1 ratio for 2 h on LB-agar supplemented with 0.4% L-arabinose. Cell suspensions were serially diluted in 1× M9 salts and plated as described above to quantify end-point cfu. Competitive indices were calculated as the final ratio of inhibitor to target cells divided by the initial ratio of inhibitor to target cells. All competitions were performed independently three times and data are presented as the mean ± standard error.

## In vitro crosslinking

Unmodified His$_6$-lipase domains (in 8 M urea) were mixed with purified Δss-Tli at 1.1:1 ratio, and the mixtures dialyzed against 20 mM sodium phosphate (pH 7.0) to allow complex formation. The resulting His$_6$-lipase•Δss-Tli complexes (17 μM) were treated with varying concentration of methylglyoxal or glyoxal at 37 °C for 3 h. Treated complexes were analyzed by SDS-PAGE. To isolate in vitro crosslinked lipases for activity assays, methylglyoxal (1 mM) and glyoxal (1 mM) treated complexes were bound to Ni$^{2+}$-NTA agarose for 90 min, then Δss-Tli

proteins were dissociated with three guanidine lysis buffer washes. His$_6$-lipase domains were then eluted from the resins with 6 M guanidine-HCl, 20 mM Tris-HCl (pH 8.0), 250 mM imidazole. The proteins were then refolded by dialysis against 20 mM Tris-HCl (pH7.5), 100 mM NaCl, quantified and used in phospholipase A1 reactions.

### Disulfide engineering, oxidation and analyses
*E. coli* CH2016 cells carrying pCH3291 were transformed with pCH7088 (wild-type), pCH8872 (Lys461Gln) or pCH1382 (Lys461Gln + disulfide), and the resulting strains were grown and induced as described above. Cell pellets were resuspended in 20 mM Tris-HCl (pH 7.5), 150 mM NaCl, 0.05% Triton X-100, 5 mM imidazole, 10 μM CuCl$_2$ and incubated with 0.1 mg/mL lysozyme for 15 min prior to sonication. Cell lysates were clarified by centrifugation at 14,462$g$ for 10 min at 4 °C, and the supernatants incubated with Ni$^{2+}$-NTA agarose for 90 min at ambient temperature. Resins were washed 5 times with 8 M urea for wild-type and Lys461Gln lipase domains, and 5 times with 3 M urea for the disulfide containing lipase. Samples were eluted in native elution buffer (20 mM Tris 7.5, 150 mM NaCl, 0.05% Triton X-100, 250 mM Imidazole). Purified His$_6$-lipase domains were dialyzed against 20 mM Tris-HCl (pH 7.5), 100 mM NaCl, quantified and assayed for phospholipase A1 activity.

### Statistics and reproducibility
Standard errors were calculated using either GraphPad Prism (version 9.4.0) or Microsoft Excel (version 16.63.1). All competition co-cultures, crosslinking experiments and phospholipase activity assays were performed independently at least three times with similar results.

### Reporting summary
Further information on research design is available in the Nature Portfolio Reporting Summary linked to this article.

## Data availability
All data generated during this study are provided within the manuscript or the Supplementary Information files. All biological materials are available to qualified researchers upon request. Coordinates for the lipase•immunity protein complex structures have been deposited with the Protein Data Bank under accession numbers 9CYS and 7UBZ. The source data underlying Figs. 1a-e, 3a-f, 4a-b, 5a-e, 6c-d and Supplementary Figs. 1a-d, 3, 4, 5a-b, 6a-b are provided as Source Data files. Source data are provided with this paper.

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

## Acknowledgements

The authors thank the Advanced Light Source at Berkeley National Laboratories and the Stanford Synchrotron Radiation Lightsource for help in data collection, and Julian Whitelegge (University of California, Los Angeles) for discussions about mass spectrometry analyses. This work was supported by National Institutes of Health grants GM117930 (CSH) and GM144437 (CWG & CSH).

## Author contributions

Conceptualization: S.J.J., C.S.H.; Methodology: S.J.J., B.J.C., F.G.S., R.d.M., C.W.G., C.S.H.; Investigation: S.J.J., B.J.C., F.G.S., C.C.H., C.S.H.; Visualization: B.J.C., C.W.G., C.S.H.; Funding acquisition: C.W.G., C.S.H.; Project administration: C.W.G., C.S.H.; Supervision: C.W.G., C.S.H.; Writing—original draft: C.S.H.; Writing—review & editing: S.J.J., B.J.C., F.G.S., C.W.G., C.S.H.

## Competing interests

The authors declare no competing interests.
