## [Transparent Peer Review file · Nature Communications]

Advanced glycation end-product crosslinking activates a type VI secretion system phospholipase effector protein

Corresponding Author: Professor Christopher Hayes

Version 0:

Reviewer comments:

Reviewer #1

(Remarks to the Author)

This manuscript "Advanced glycation end-product (AGE) crosslinking as a bona fide post-translational modification" describes the activation of an antibacterial phospholipase by the formation of an advanced glycation end-product (AGE) crosslink MODIC. Overall, this is a well written study with some compelling data, including a crystal structure of the Tle lipase containing an intramolecular MODIC crosslink. This study is significant as it provides support for a newly emerging paradigm in the field that sees glycation as not only random damage that develops over time, but also as a functional protein modification. They also show that a nearby glutamic acid stabilizes the formation of the MODIC crosslink through hydrogen bonding. We therefore find the results of this paper quite exciting not only from the perspective of glycation, but also as it relates to the growing number of examples in which bacteria use novel PTMs to influence infection and/or immunity. At the same time, we also had some significant concerns about certain aspects of the study, most notably the mass spectrometry and experiments to rule out catalysis by Tli, which must be addressed with additional experiments. We also have some additional clarifications and suggestions that should also be addressed. Thus, we expect that this article will become suitable for publication in Nature Communications after major revisions.

1. One of our most major concerns is regarding the mass spectrometry data presented. The MS data presented is not high resolution (<5 ppm). The error (ppm) reported in Figure 1 is quite far off (around 60 ppm for the mutant versions) which is simply not high resolution enough to be convincing. I believe the x axis is also mislabeled as those appear to be deconvoluted spectra. Additionally, reporting a 1000 ppm (!?) for the expected "wild-type" modified" is confusing – it would be better to show it for the expected mass of the modified WT protein. This data would also be strengthened significantly with comparisons by MS to the unmodified Tle. A Xevo G2-XS QTOF should be capable of performing high resolution mass spectrometry data on intact proteins that should be accurate within 1 amu (or less).
2. Similarly, Fig 1 describes a mass change of +36 amu but Extended Fig 1 describes a mass change of +33 amu for the MODIC modification. As per the earlier point, this is not high resolution enough and needs to be repeated. Additionally, considering the importance of the protein modification, additional characterization should be done by MS/MS (also possible on the Xevo QTOF used) to confirm the exact mass of the crosslinked fragment and the sites of crosslinking.
3. Another major concern is the characterization of the +36 mass shift as a genuine AGE. Line 127 states that "Mass spectrometry analysis also shows that the Arg180Lys and Lys461Gln domain variants 128 do not react with methylglyoxal (Fig. 1e)" but I do not see data showing that in Figure 1. Instead I see that they were unmodified after they were expressed from bacteria (unlike wild-type). These statements would be significantly strengthened by treatments in vitro with MGO or GO leading to no reaction (or more likely, no crosslinking) with those proteins.
4. Similarly, we also do not see data in which MGO concentrations in ECL are measured or modulated in any way, both of which could influence MODIC formation. For example, if Glo/II were upregulated, would you see less crosslinking? Or vice versa?
5. Additionally, for the experiments that did evaluate MGO addition in vitro, a greater range of MGO concentrations (including lower MGO concentrations – 10 equivalents was the lowest tested in this study) and time course studies should also be done to evaluate MODIC formation.

6. Most concerning, the authors have not ruled out the possibility that Tli is catalyzing AGE formation. As they state in the conclusion “This crosslinking must be extraordinarily efficient, because T6SS effectors are deployed within minutes of synthesis” – while not entirely unheard of (e.g. <https://www.nature.com/articles/s41586-018-0622-0>), there are scant examples of scenarios where non-enzymatic glycation events are so rapid. Further stress testing of this hypothesis is needed, ideally including studies that evaluate intramolecular Tle crosslinking in the absence of Tli.

7. The disulfide crosslinking experiments are interesting, but it is unfortunate that the Cys substitutions were made at different positions compared to where MODIC is formed. A similar approach should be taken using the same reactive sites.

8. The writing in the introduction is quite dense and may be difficult to follow for those outside of the bacterial pathogenesis community. It would be helpful if the authors could include some cartoons or schemes that better explain their system and readers can use to help understand the functional consequence of the MODIC crosslink.

9. Similarly, we also notice that term “modification” is used somewhat inconsistently and prematurely in the manuscript. As an example in line 61 “Fortunately, the C-terminal lipase domain of Tle (residues Thr172 - Ala472) is also subject to modification when produced with Δ ss-Tli” – at this point, a protein modification had not yet been detected so it would be better to describe the Tle lipase domain as also exhibiting increased mobility on gel and/or running with an anomalous mass shift by gel electrophoresis. There are many more similar examples throughout the paper, and addressing these would greatly aid with clarity.

10. The title and abstract seem to focus solely on glycation and much less so on the functional activation of Tle and the significance of this activation. We respectfully suggest that the authors may want to consider an alternative title and abstract that more clearly captures the entirety of this study. An example would be “An AGE crosslink activates a bacterial phospholipase immunity protein”, but we are open to whatever the authors prefer.

Reviewer #2

(Remarks to the Author)

Reviewer #3

(Remarks to the Author)

RE: Advanced glycation end-product (AGE) crosslinking as a bona fide post-translational modification by Jensen and colleagues

In the underlying manuscript the authors identified and characterized a novel bacterial post-translation modification event involving the antibacterial phospholipase (Tle) from *Enterobacter cloacae* (ECL), a substrate of the Type VI secretion system (T6SS), that is dependent on the interaction with the T6SS lipase immunity (Tli) protein. Using a combination of different expression systems in *E. coli* (arabinose and IPTG inducible), the generation of a truncated (residues Thr172 - Ala472 of C-terminal lipase domain) and His6-tagged version of Tle in combination with an exclusively cytoplasmic version of Tli (by deleting the signal peptide sequence = Δ ss-Tli), the authors demonstrate that the lipase domain of Tle is only modified when synthesized in combination with Tli, and that this modification is correlated with enzymatic activity.

Based on the observed discrepancies between theoretical mass/PAGE running behavior and mass spectrometric analyses, the authors subsequently solved the crystal structure of the His6-lipase- Δ ss-Tli complex. They found that the protein interface is extensive and that helix α 6 that is covering the active site in the absence of Tli is being pulled away through interactions with Δ ss-Tli, preventing interaction with lipid substrates and thus neutralizing Tle. The authors also observed that Arg180 from helix α 1 and Lys461 in helix α 13 are covalently linked joining the N- and C-termini of the lipase domain into a topological circle. This Arg-Lys crosslink (known to occur after prolonged exposure to glyoxal or methylglyoxal) is consistent with a methylglyoxal-derived imidazolium crosslink (MODIC). In vitro assays and in vivo activity/complementation analyses in ECL further revealed that Arg180Lys and Lys461Gln substitutions prevent inhibition concluding that AGE crosslinking is indeed required for phospholipase activity. In addition, intramolecular crosslinking of purified His6-lipase with methylglyoxal (and with glyoxal at lower efficiency) could only be achieved in the presence of Δ ss-Tli (no crosslinking with Arg180Lys and Lys461Gln lipase variants or in the absence of Tli) and promotes (when purified away from Tli) Tlp phospholipase A1 activity. Moreover, crosslinked Tle could again be neutralized with Δ ssTli. Lastly, by engineering a disulfide bond to the Lys461Gln domain, the authors demonstrate restoration of phospholipase activity that could be neutralized with Δ ss-Tli in vitro but not in vivo (since no disulfide bond do not form in the cytoplasm).

In general: the manuscript is very well written and the results are presented logically and in a convincing manner; every experimental design has the proper controls and I specifically liked: i) the in vivo versus in vitro comparisons including the elegant *E. coli* expression system demonstrating that Tle modification and Type-6 secretion are two independent processes, and ii) the engineering of the disulfide bond and the demonstration that Tlp activity could only be restored under non-reducing (in vitro) conditions, but not in the ECL cytoplasm emphasizing the utilization of the AGE crosslink in lieu of a conventional disulfide bond to promote protein function under reducing conditions. In addition, the authors discuss their observations and results based on other documented systems; the figures and tables are very well presented and support the data/results very well. For additional information, the authors included all the necessary details in the Extended Data section.

In summary: the observed reactivity of the Tle-Tli complex with methylglyoxal is (as the authors state) indeed unprecedented and describes a novel mechanism/modification system that is used to activate proteins rather than being invariably deleterious to protein function.

Minor:

Immunoblotting Fig 1a: In the absence of Tli it is described that there is only one form of Tle. However, in Fig 1a lane 3 there seems to be some modified Tle? – or is that a degradation product? There are also lower MW bands visible around the 50 kDa range in the Extended Data Fig 5, lanes 1 & 2?

There is one structural figure that uses the 3-letter code rather than the 1 letter code – this should be consistent.

Reviewer #4

(Remarks to the Author)

In this paper, Jensen et al show that glycation is used in bacteria to cross-link an arginine-lysine pair found in a secreted lipase enzyme. Remarkably and perhaps somewhat paradoxically, the formation of this cross-link is induced through the physical interaction of the lipase with its antitoxin protein, Tli. Using a series of genetic and biochemical experiments, the authors show that this cross-link is essential for lipase function and that mutations that disrupt its formation can be overcome through the engineering of a non-native disulfide that accomplishes a similar stabilizing effect. Overall, the experiments in this study were well controlled and performed with a high degree of rigor. Moreover, the manuscript itself is superbly written. I have only minor queries for the authors' consideration.

- The molecular details of "A1 and A2 reactions" should be better explained for a non-specialist audience
- Line 80: typo "surrounded"
- Given the apparent significance of the dynamic nature of the alpha-6 helix of the lipase, a side-by-side figure of the AF2 model of the lipase alone next to the X-ray structure of the complex in Figure 2 (rather than just in the supplement) would make it easier for the reader to follow the text description on lines 90-105.
- Line 162: "To determine whether the in vitro crosslinked lipases are active, we purified the treated domains away from Δ ss-Tli and assayed their enzymatic activities". How were they purified away from Tli? Please explain methodology more clearly.
- Figure 1a and 3d: unclear from the figure or legend what antibody is being used for the immunoblots
- Figure 1a and elsewhere: For this reader, using the designation of "ECL" rather than *E. cloacae* is unnecessarily confusing.
- Overall, all of the main figure legends would benefit from additional experimental detail.
- Figure 2d: color scheme for surface potential is not defined.
- I recommend increasing the font sizes for all graphs, gel labels, activity assay labels, etc. in the Figures.

Version 1:

Reviewer comments:

Reviewer #1

(Remarks to the Author)

The revised manuscript has addressed all of my concerns from the prior submission. The authors provide clear and convincing evidence that an AGE crosslink activates a phospholipase effector, which is not only interesting both in terms of the biological function, but also points to AGE formation as a true PTM that has specific biological consequences. Overall, this is a very exciting paper that is now suitable for publication in Nature Communications.

Reviewer #2

(Remarks to the Author)

Reviewer #3

(Remarks to the Author)

Based on the review of the initial version of the manuscript there were no major concerns. The minor concerns have all been addressed by the authors. I also noticed that the paper went through a more extensive revision process in response to comments from the other reviewers and the additional information and experiments that have been included further strengthen the previously presented data and conclusions. I have no further comments or concerns.

Reviewer #4

(Remarks to the Author)

The author's have satisfactorily addressed all my concerns.

Reviewer #1 (Remarks to the Author):

This manuscript “Advanced glycation end-product (AGE) crosslinking as a bona fide post-translational modification” describes the activation of an antibacterial phospholipase by the formation of an advanced glycation end-product (AGE) crosslink MODIC. Overall, this is a well written study with some compelling data, including a crystal structure of the Tle lipase containing an intramolecular MODIC crosslink. This study is significant as it provides support for a newly emerging paradigm in the field that sees glycation as not only random damage that develops over time, but also as a functional protein modification. They also show that a nearby glutamic acid stabilizes the formation of the MODIC crosslink through hydrogen bonding. We therefore find the results of this paper quite exciting not only from the perspective of glycation, but also as it relates to the growing number of examples in which bacteria use novel PTMs to influence infection and/or immunity. At the same time, we also had some significant concerns about certain aspects of the study, most notably the mass spectrometry and experiments to rule out catalysis by Tli, which must be addressed with additional experiments. We also have some additional clarifications and suggestions that should also be addressed. Thus, we expect that this article will become suitable for publication in Nature Communications after major revisions.

1. One of our most major concerns is regarding the mass spectrometry data presented. The MS data presented is not high resolution (<5 ppm). The error (ppm) reported in Figure 1 is quite far off (around 60 ppm for the mutant versions) which is simply not high resolution enough to be convincing. I believe the x axis is also mislabeled as those appear to be deconvoluted spectra. Additionally, reporting a 1000 ppm (!?) for the expected “wild-type” modified is confusing – it would be better to show it for the expected mass of the modified WT protein. This data would also be strengthened significantly with comparisons by MS to the unmodified Tle. A Xevo G2-XS QToF should be capable of performing high resolution mass spectrometry data on intact proteins that should be accurate within 1 amu (or less).

We apologize for the confusion surrounding the mass spectrometry analysis. In the revision, we have more clearly explained that the mass of modified His₆-Tle is 36 Da greater than expected, which excludes proteolytic processing as an explanation for the gel mobility shift (lines 104 - 106). We also note in the revision that unmodified Tle/lipase domain does not ionize well for mass spectrometry. Therefore, we have been unable to measure the mass of unmodified lipase. This phenomenon also affects analysis of the Arg180Lys and Lys461Gln variants as they cannot undergo crosslinking. The mutant mass data we have is from purified His₆-lipase•Δss-Tli complexes, which we show as complete deconvoluted spectra (Figs. 3b & 3c) in the revision. Finally, we have reprocessed the mass spectrometry data and values are now generally within 1 Da of predicted masses.

2. Similarly, Fig 1 describes a mass change of +36 amu but Extended Fig 1 describes a mass change of +33 amu for the MODIC modification. As per the earlier point, this is not high resolution enough and needs to be repeated. Additionally, considering the importance of the protein modification, additional characterization should be done by MS/MS (also possible on the Xevo QTOF used) to confirm the exact mass of the crosslinked fragment and the sites of crosslinking.

The reprocessed mass spectrometry data indicate a ~36 Da increase for both His₆-tagged lipase domain (Fig. 1c) and His₆-tagged full-length Tle (Supplementary Fig. 1b). We have chosen not to pursue MS/MS analysis because our manuscript includes two high-resolution crystal structures, both of which show continuous electron density between the side chains of

Arg180 and Lys461. We also provide site-directed mutagenesis data and *in vitro* crosslinking experiments that collectively support our conclusion that methylglyoxal crosslinks the side-chains of Arg180 and Lys461.

3. Another major concern is the characterization of the +36 mass shift as a genuine AGE. Line 127 states that “Mass spectrometry analysis also shows that the Arg180Lys and Lys461Gln domain variants 128 do not react with methylglyoxal (Fig. 1e)” but I do not see data showing that in Figure 1. Instead I see that they were unmodified after they were expressed from bacteria (unlike wild-type). These statements would be significantly strengthened by treatments *in vitro* with MGO or GO leading to no reaction (or more likely, no crosslinking) with those proteins.

The original manuscript showed that Arg180Lys and Lys461Gln variants are not subject to *in vitro* methylglyoxal crosslinking in Figs. 4c & 4d. In the revision, these data are now presented as Figs. 5c & 5d.

4. Similarly, we also do not see data in which MGO concentrations in ECL are measured or modulated in any way, both of which could influence MODIC formation. For example, if Glol/II were upregulated, would you see less crosslinking? Or vice versa?

The revision includes experiments that attempt to modulate methylglyoxal concentrations in both *E. coli* and *E. cloacae* cells. Supplementary Fig. 3 shows that lipase domain crosslinking is not appreciably affected in *E. coli* Δ *mgsA* cells (which lack methylglyoxal synthase). We also treated *E. coli* cells with high concentrations of aminoguanidine, which scavenges methylglyoxal, and saw no effect on crosslinking. However, overexpression of methylglyoxal detoxification enzymes has some effect on crosslinking in *E. coli* cells (Fig. 4a) and a moderate effect on Tle-mediated competition in *E. cloacae* cells (Fig. 4b). These new experiments are described on lines 535 - 558 of the revision.

5. Additionally, for the experiments that did evaluate MGO addition *in vitro*, a greater range of MGO concentrations (including lower MGO concentrations – 10 equivalents was the lowest tested in this study) and time course studies should also be done to evaluate MODIC formation.

Given that no *in vitro* crosslinking is observed at the lowest methylglyoxal concentration (1 μ M) and crosslinking is virtually complete at the highest concentration (5 mM), it is not clear what will be learned by testing concentrations outside of the already presented range. A new time course experiment at 1 mM methylglyoxal is included in the revision as Supplementary Fig. 4.

6. Most concerning, the authors have not ruled out the possibility that Tli is catalyzing AGE formation. As they state in the conclusion “. This crosslinking must be extraordinarily efficient, because T6SS effectors are deployed within minutes of synthesis” – while not entirely unheard of (e.g. <https://www.nature.com/articles/s41586-018-0622-0>), there are scant examples of scenarios where non-enzymatic glycation events are so rapid. Further stress testing of this hypothesis is needed, ideally including studies that evaluate intramolecular Tle crosslinking in the absence of Tli.

We haven't considered Tli-mediated catalysis because the immunity protein is ~33 Å from the crosslink site, which appears inconsistent with direct enzymatic catalysis. We favor a model in which Tli promotes Tle folding to position helices α 1 and α 13 for spontaneous reaction. We also note that the original manuscript showed that the lipase does not undergo *in vitro* crosslinking in the absence of immunity protein. These data are now in Supplementary Fig. 5a of the revision.

7. The disulfide crosslinking experiments are interesting, but it is unfortunate that the Cys substitutions were made at different positions compared to where MODIC is formed. A similar approach should be taken using the same reactive sites.

Disulfide formation requires specific bond angles, and therefore the suggested experiment is unlikely to work as anticipated by the reviewers. We used crystal structure coordinates in combination with Disulfide-by-design algorithm to identify all Cys substitution pairs with the potential to link helices $\alpha 1$ and $\alpha 13$. The output did not include Cys substitutions for positions 180 or 461, and AlphaFold2 modeling predicts that such a variant would not form an intramolecular disulfide.

8. The writing in the introduction is quite dense and may be difficult to follow for those outside of the bacterial pathogenesis community. It would be helpful if the authors could include some cartoons or schemes that better explain their system and readers can use to help understand the functional consequence of the MODIC crosslink.

This is an excellent suggestion. We have added a new figure (Fig. 7) that outlines our model for MODIC activation in the context of type 6 secretion.

9. Similarly, we also notice that term “modification” is used somewhat inconsistently and prematurely in the manuscript. As an example in line 61 “Fortunately, the C-terminal lipase domain of Tle (residues Thr172 - Ala472) is also subject to modification when produced with $\Delta ss-Tli$ ” – at this point, a protein modification had not yet been detected so it would be better to describe the Tle lipase domain as also exhibiting increased mobility on gel and/or running with an anomalous mass shift by gel electrophoresis. There are many more similar examples throughout the paper, and addressing these would greatly aid with clarity.

We have revised the opening results section to clarify the use of "modification".

10. The title and abstract seem to focus solely on glycation and much less so on the functional activation of Tle and the significance of this activation. We respectfully suggest that the authors may want to consider an alternative title and abstract that more clearly captures the entirety of this study. An example would be “An AGE crosslink activates a bacterial phospholipase immunity protein”, but we are open to whatever the authors prefer.

We have taken the reviewers' suggestion and changed the manuscript title to emphasize the biological effects on T6SS.

Reviewer #2 (Remarks to the Author):

Reviewer #3 (Remarks to the Author):

RE: Advanced glycation end-product (AGE) crosslinking as a bona fide post-translational modification by Jensen and colleagues

In the underlying manuscript the authors identified and characterized a novel bacterial post-translation modification event involving the antibacterial phospholipase (Tle) from *Enterobacter cloacae* (ECL), a substrate of the Type VI secretion system (T6SS), that is dependent on the interaction with the T6SS lipase immunity (Tli) protein. Using a combination of different expression systems in *E. coli* (arabinose and IPTG inducible), the generation of a truncated (residues Thr172 - Ala472 of C-terminal lipase domain) and His6-tagged version of Tle in combination with an exclusively cytoplasmic version of Tli (by deleting the signal peptide sequence = Δ ss-Tli), the authors demonstrate that the lipase domain of Tle is only modified when synthesized in combination with Tli, and that this modification is correlated with enzymatic activity.

Based on the observed discrepancies between theoretical mass/PAGE running behavior and mass spectrometric analyses, the authors subsequently solved the crystal structure of the His6-lipase- Δ ss-Tli complex. They found that the protein interface is extensive and that helix α 6 that is covering the active site in the absence of Tli is being pulled away through interactions with Δ ss-Tli, preventing interaction with lipid substrates and thus neutralizing Tle. The authors also observed that Arg180 from helix α 1 and Lys461 in helix α 13 are covalently linked joining the N- and C-termini of the lipase domain into a topological circle. This Arg-Lys crosslink (known to occur after prolonged exposure to glyoxal or methylglyoxal) is consistent with a methylglyoxal-derived imidazolium crosslink (MODIC). In vitro assays and in vivo activity/complementation analyses in ECL further revealed that Arg180Lys and Lys461Gln substitutions prevent inhibition concluding that AGE crosslinking is indeed required for phospholipase activity. In addition, intramolecular crosslinking of purified His6-lipase with methylglyoxal (and with glyoxal at lower efficiency) could only be achieved in the presence of Δ ss-Tli (no crosslinking with Arg180Lys and Lys461Gln lipase variants or in the absence of Tli) and promotes (when purified away from Tli) Tlp phospholipase A1 activity. Moreover, crosslinked Tle could again be neutralized with Δ ssTli. Lastly, by engineering a disulfide bond to the Lys461Gln domain, the authors demonstrate restoration of phospholipase activity that could be neutralized with Δ ss-Tli in vitro but not in vivo (since no disulfide bond do not form in the cytoplasm).

In general: the manuscript is very well written and the results are presented logically and in a convincing manner; every experimental design has the proper controls and I specifically liked: i) the in vivo versus in vitro comparisons including the elegant *E. coli* expression system demonstrating that Tle modification and Type-6 secretion are two independent processes, and ii) the engineering of the disulfide bond and the demonstration that Tlp activity could only be restored under non-reducing (in vitro) conditions, but not in the ECL cytoplasm emphasizing the utilization of the AGE crosslink in lieu of a conventional disulfide bond to promote protein function under reducing conditions. In addition, the authors discuss their observations and results based on other documented systems; the figures and tables are very well presented and support the data/results very well. For additional information, the authors included all the necessary details in the Extended Data section.

In summary: the observed reactivity of the Tle-Tli complex with methylglyoxal is (as the authors state) indeed unprecedented and describes a novel mechanism/modification system that is used to activate proteins rather than being invariably deleterious to protein function.

Minor:

Immunoblotting Fig 1a: In the absence of Tli it is described that there is only one form of Tle. However, in Fig 1a lane 3 there seems to be some modified Tle? – or is that a degradation product? There are also lower MW bands visible around the 50 kDa range in the Extended Data

Fig 5, lanes 1 & 2?

We believe that these bands (which are also apparent with the R180K and K461Q variants in Fig. 3f) are degradation products. Given that unmodified full-length Tle is insoluble, perhaps it is more susceptible to degradation. Moreover, because active Tle is not deployed from *E. cloacae* Δtli cells, it is unlikely that these products correspond to AGE crosslinked Tle.

There is one structural figure that uses the 3-letter code rather than the 1 letter code – this should be consistent.

We used three-letter code to indicate the catalytic triad residues of overlaid lipase domains. Because these labels lack numerical descriptors, we felt that it would clearer to use three-letter code rather than isolated "S", "H" and "D/E" labels.

Reviewer #4 (Remarks to the Author):

In this paper, Jensen et al show that glycation is used in bacteria to cross-link an arginine-lysine pair found in a secreted lipase enzyme. Remarkably and perhaps somewhat paradoxically, the formation of this cross-link is induced through the physical interaction of the lipase with its antitoxin protein, Tli. Using a series of genetic and biochemical experiments, the authors show that this cross-link is essential for lipase function and that mutations that disrupt its formation can be overcome through the engineering of a non-native disulfide that accomplishes a similar stabilizing effect. Overall, the experiments in this study were well controlled and performed with a high degree of rigor. Moreover, the manuscript itself is superbly written. I have only minor queries for the authors' consideration.

- The molecular details of "A1 and A2 reactions" should be better explained for a non-specialist audience

We now indicate that A1 and A2 reactions entail hydrolysis of acyl chains from the *sn*-1 and *sn*-2 positions of phospholipids.

- Line 80: typo "surrounded"

Error corrected

- Given the apparent significance of the dynamic nature of the alpha-6 helix of the lipase, a side-by-side figure of the AF2 model of the lipase alone next to the X-ray structure of the complex in Figure 2 (rather than just in the supplement) would make it easier for the reader to follow the text description on lines 90-105.

We have integrated the AlphaFold2 comparison into Fig. 2 as suggested.

- Line 162: "To determine whether the in vitro crosslinked lipases are active, we purified the treated domains away from Δss -Tli and assayed their enzymatic activities". How were they purified away from Tli? Please explain methodology more clearly.

The revision now states that the crosslinked His₆-lipases were purified by Ni²⁺-affinity chromatography under denaturing conditions followed by dialysis to refold the enzymes.

- Figure 1a and 3d: unclear from the figure or legend what antibody is being used for the immunoblots

We now indicate that anti-Tle antibodies were used with labels in the figures and descriptions in the legends

- Figure 1a and elsewhere: For this reader, using the designation of "ECL" rather than *E. cloacae* is unnecessarily confusing.

We agree, ECL has been replaced with *E. cloacae* throughout the revision.

- Overall, all of the main figure legends would benefit from additional experimental detail.
- Figure 2d: color scheme for surface potential is not defined.

The figure legends have all been modified to briefly outline experimental procedures.

- I recommend increasing the font sizes for all graphs, gel labels, activity assay labels, etc. in the Figures.

We have increased the font size for the smaller labels. We note that Journal art guidelines state that figures fonts should be a maximum of 7 pt.